# Estimating Urban Evapotranspiration at 10m Resolution Using Vegetation Information from Sentinel-2: A Case Study for the Beijing Sponge City

Xuanze Zhang *  and Peilin Song 

Key Laboratory of Water Cycle and Related Land Surface Processes, Institute of Geographic Sciences and Natural Resources Research, Chinese Academy of Sciences, Beijing 100101, China; songpl@igsnrr.ac.cn
* Correspondence: xuanzezhang@igsnrr.ac.cn

**Abstract:** Estimating accurately evapotranspiration (ET) in urban ecosystems is difficult due to the complex surface conditions and a lack of fine measurement of vegetation dynamics. To overcome such difficulties using recent developments of remote sensing technology, we estimate leaf area index (LAI) from Sentinel-2-based Normalized Difference Vegetation Index (NDVI) using the NDVI–LAI nonlinear relationship. By applying Sentinel-2-based LAI and land cover classification (LCC) to a carbon-water coupling model (PML-V2.1) with surface meteorological forcing data as input, we, for the first time, estimate monthly ET at 10m × 10m resolution for the Beijing Sponge City. Results show that for the whole sponge city during June 2018, the LAI, ET and gross primary productivity (GPP) are 0.83 $m^2$ $m^{-2}$, 1.6 mm $d^{-1}$ and 2.8 gC $m^{-2}$ $d^{-1}$, respectively. For different LCCs, lakes and rivers have the highest ET ($\geq$8 mm $d^{-1}$), followed by mixed forests and croplands (ET is 4–6 mm $d^{-1}$ and LAI is 2–3 $m^2$ $m^{-2}$) with dominant contribution (>80%) from plant transpiration, while grasslands (2–4 mm $d^{-1}$) have 50–70% from transpiration due to smaller LAI (1~2 $m^2$ $m^{-2}$). The impervious surfaces occupying ~60% of the sponge city area, have the smallest ET (<2.0 mm $d^{-1}$) in which interception evaporation by impervious surface contributes 20–30%, and transpiration from greenbelts (0.5–1.0 $m^2$ $m^{-2}$ of LAI) contributes 40–50%. These findings can provide a valuable scientific basis for policymaking and urban water use planning. This study proposes a Sentinel-2-based technology for estimating ET as a feasible framework to evaluate city-level hydrological dynamics in urban ecosystems.

**Keywords:** evaporation; evapotranspiration; LAI; NDVI; urban ecosystem; sponge city; PML-V2; Penman–Monteith equation; Sentinel-2

## 1. Introduction

Owing to the high heterogeneity and complexity in urban ecosystems, it is rather difficult to monitor or predict the hydrological dynamics of urban surfaces [1]. Some megacities, e.g., Beijing—the capital city of China—have experienced strong urbanization, large population inflow, island effect and climate change during the past few decades [2]. These changes induce urban hydrological processes to be highly uncertain and make policymakers face tough challenges in water use planning and management. Therefore, there is an urgent need to accurately estimate urban hydrological processes.

Evapotranspiration (ET), as a key component of the urban hydrological processes and surface energy balance, plays an important role in regulating water resource supply and relieving the urban island effect (e.g., surface cooling) [3]. Different from natural ecosystems, the urban ecosystems include large proportions of artificial modifications in land cover, such as impervious surfaces including roofs, squares and cement or asphalt roads. These man-made reconstructions could contribute a large fraction of evaporation [4–6], but the quantification at city levels remains highly uncertain due to a lack of clearly distinguishing estimations of ET between impervious surfaces and vegetated or bare-soil lands. The

good news is that recent developments of fine resolution remote sensing for land use and land cover classification, vegetation dynamics and environmental monitoring provide new opportunities to estimate urban ET more accurately [7]. For example, the Sentinel-2, as part of the European Commission's Copernicus program with the launch of satellite Sentinel-2A on 23 June 2015, are monitoring variability in global land surface conditions at a 10–60m resolution and a 5–10-day revisit [8].

In this study, we take the sponge city project in Beijing city as a study case to estimate ET of the urban ecosystem at 10m resolution using the satellite-based land cover map and vegetation information derived from Sentinel-2 data. Beijing city has 5-year mean annual precipitation of 560 mm and mean annual temperature of 12 °C, with the potential evaporation of 550~600 mm year$^{-1}$. To reduce stress on water supply (e.g., ~30 m$^3$ per capita water use per year in Beijing) and urban environment, the Beijing Sponge City project was started on 4 December 2017, aiming at turning 20% of Beijing city into a sponge city covered area by 2020 [9]. Therefore, to evaluate the benefit of this project, it is essential to implement a city-level assessment of the project-induced ecohydrological changes at fine resolution.

## 2. Materials and Methods

### 2.1. Observational Forcing Datasets

#### 2.1.1. Land Cover Map at 10m Resolution Derived from Sentinel-2

The land cover classification (LCC) global map at 10m resolution was obtained from FROM-GLC10 [7]. The FROM-GLC10 LCC data is developed based on Sentinel-2 data in 2017 with Google Earth Engine, and the overall accuracy of this LCC validated against the circa 2015 validation sample is 73% [7]. The LCC data includes 10 classes (i.e., cropland, forest, grassland, shrubland, wetland, tundra, impervious surface, bare land, and snow/ice). The most advances of the FROM-GLC10 LCC map compared to previous Landsat series-based LCC products are that it provides more spatial detail, better distinguish the forest from shrub or grassland classes, and better performance in coastal areas [7].

#### 2.1.2. NDVI and LAI at 10m Resolution Derived from Sentinel-2

We calculate the Normalized Difference Vegetation Index (NDVI) from the Sentinel-2 reflected radiance by

$$NDVI = \frac{R_{nir} - R_{red}}{R_{nir} + R_{red}} \tag{1}$$

where $R_{nir}$ and $R_{red}$ are the spectral bands at near infrared (842 nm) and red (665 nm), respectively. The Sentinel-2 reflectance data are available at the USGS EROS Center (https://www.usgs.gov/centers/eros, accessed on 8 April 2021). The leaf area index (LAI) at a 10m resolution was derived from the retrieved Sentinel-2 NDVI using a nonlinear regression model between LAI and NDVI,

$$LAI = a * exp(b * NDVI) + c \tag{2}$$

where the parameters *a*, *b* and *c* are determined as 12.4, 6.4 and 0.6, respectively.

The determination process was based on MODIS-based NDVI and LAI products, which was described as: (i) The MODIS LAI (MOD15A2H) and surface reflectance (SR) products (MOD09A1) at 500m were collected over the study area (Beijing Sponge City) for the summer months (June, July, and August) from 2013 to 2019. The NDVI was then estimated using the 500m SR product (Equation (1)). (ii) MODIS LAI and NDVI values were collocated on the pixel basis. As the MODIS LAI product has a valid range between 0 and 6.9, with a precision of 0.1, we classified all NDVI values into 69 groups based on unique LAI values (eliminating the zero-LAI group). The probability density plots for each group are shown in Figure 1a. (iii) For each LAI-based value group, the probability density was fitted using the Gaussian distribution function. Then, the NDVI value corresponded by the maximum probability density was extracted and collocated with the specific LAI value.



The result shows a strong exponential relationship between NDVI and LAI, especially when the LAI value increased beyond 0.5–0.6 (Figure 1b). Therefore, the scatter values were fitted using the exponential model (Equation (2)), which resulted in an $R^2$ of 0.82, implying that such exponential model in Equation (2) is robust for the Beijing Sponge City.

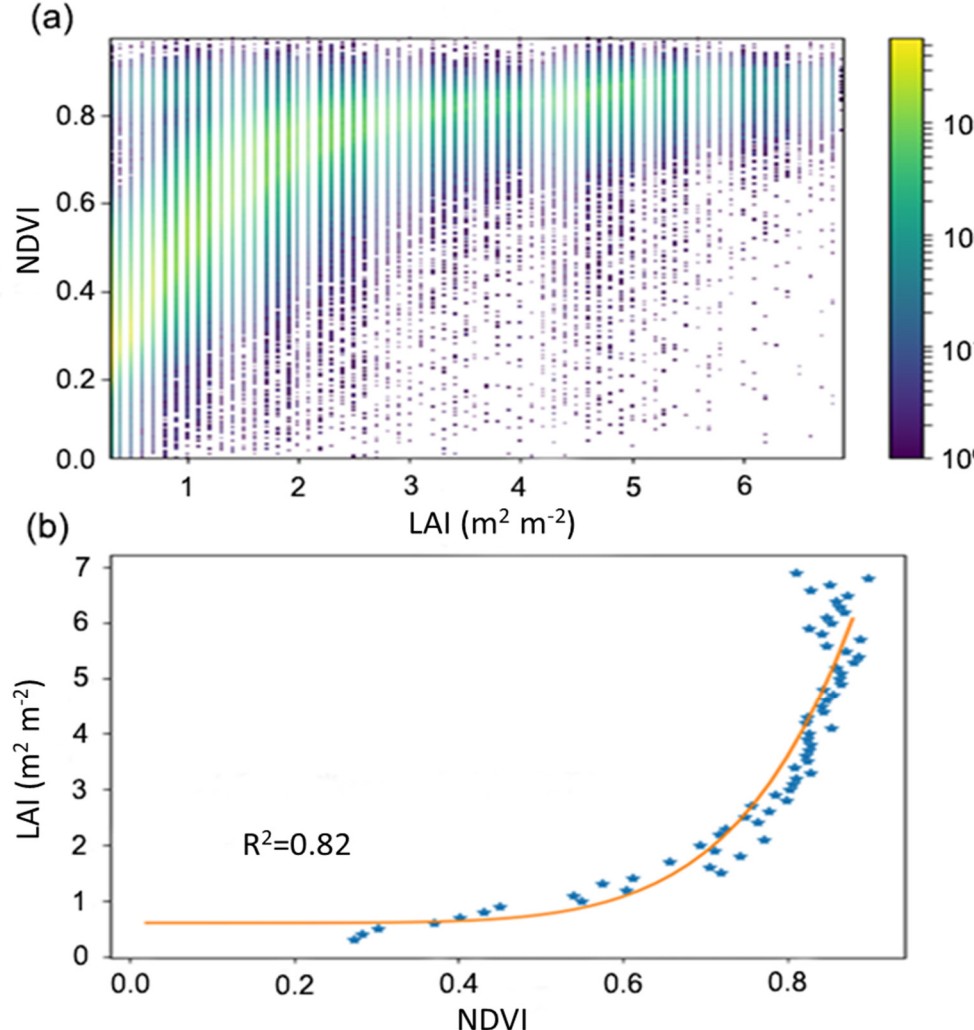

**Figure 1.** The nonlinear relationship between LAI and NDVI. (**a**) Frequency distribution of NDVI–LAI. Higher values of the scatter number in (**a**) indicate stronger relation between LAI and NDVI. (**b**) Regression between LAI and NDVI. The brown curve was fitted using Equation (2) from the NDVI–LAI values (blue stars) under the maximum probability density.

### 2.1.3. Surface Climate Driving Dataset

To estimate evapotranspiration of urban ecosystem at 10m resolution, a high-resolution surface climate forcing data including precipitation, surface air temperature, wind speed, surface pressure, specific humidity, downward shortwave and longwave radiations, etc., is needed to drive the terrestrial evapotranspiration model (PML-V2 model, see Section 2.2). In this study, we used the China Meteorological Forcing Dataset (CMFD) version 1 at $0.1° \times 0.1°$ and daily resolution for June 2018 as input for the PML-V2 model. The CMFD V1.0 dataset covered the period of 1979–2018 and was downscaled from station-based data, TRMM satellite-based precipitation, GEWEX-SRB shortwave radiation and the GLDAS forcing dataset [10]. The surface climate driving variables used for the Beijing Sponge City area were spatially bilinearly interpolated onto a 10m × 10m resolution. The monthly $CO_2$ concentration observed in June 2018 is set as 407 ppm.

## 2.2. PML-V2.1 Model

Version 2 of the Penman–Monteith–Leuning model (PML-V2) was developed by coupling the widely-used photosynthesis model [11] and a canopy stomatal conductance model [12] with the Penman–Monteith energy balance equation [13] to jointly estimate gross primary productivity (GPP), $E_c$ and $E_s$ [14–18]. The PML-V2 model also simulates the $E_i$ based on a revised Gash-model scheme [19]. The PML-V2 model has been applied to successfully produce the MODIS LAI-based global GPP and ET products at a 500m and 8-day resolution from 2002 to present, which were noticeably better than most widely used GPP and ET products [16]. In this study, we incorporated modules of impervious surface evaporation ($E_u$) and open-water evaporation ($E_w$) into the PML model (PML-V2.1) to make it suitable for urban ecosystems. Key parameters used in the PML-V2.1 model are provided in Table 1. The following shows the detailed description for PML-V2.1.

**Table 1.** Key parameters used in the PML-V2.1 model.

| Parameter | Definition | Unit | Land Cover Classification [a] | | | | | | | |
|---|---|---|---|---|---|---|---|---|---|---|
| | | | CRO | MIF | GRA | SHR | WET | WAT | IMP | BAR |
| $\alpha$ | Surface albedo for shortwave radiation | – | 0.150 | 0.150 | 0.250 | 0.250 | 0.250 | 0.050 | 0.350 | 0.350 |
| $\varepsilon$ | Emissivity for longwave radiation | – | 0.960 | 0.990 | 0.950 | 0.950 | 0.960 | 0.990 | 0.940 | 0.940 |
| $D_0$ | Reference vapor pressure deficit at stomatal conductance reduction | kPa | 2.000 | 0.552 | 0.638 | 0.864 | 0.661 | 0.700 | 0.552 | 0.864 |
| $k_Q$ | Extinction coefficient of PAR | – | 0.721 | 0.386 | 0.595 | 0.230 | 0.996 | 0.600 | 0.386 | 0.230 |
| $k_A$ | Extinction coefficient of available energy | – | 0.899 | 0.899 | 0.900 | 0.888 | 0.888 | 0.700 | 0.899 | 0.888 |
| $S_{leaf}$ | Specific canopy rainfall storage capacity per unit leaf area | mm | 0.010 | 0.198 | 0.227 | 0.014 | 0.022 | 0.000 | 0.198 | 0.014 |
| $F_{ER0}$ | Specific ratio of evaporation rate over rainfall intensity per unit vegetation cover | – | 0.092 | 0.256 | 0.010 | 0.010 | 0.017 | 0.000 | 0.256 | 0.010 |
| $S_U$ | Specific canopy rainfall storage capacity per unit impervious surface area | mm | 0.014 | 0.014 | 0.014 | 0.014 | 0.014 | 0.014 | 0.014 | 0.014 |
| $LAI_{ref}$ | Reference LAI | m | 5.000 | 5.000 | 5.000 | 5.000 | 5.000 | 5.000 | 5.000 | 5.000 |
| $h$ | Canopy height | m | 1.000 | 10.00 | 0.5000 | 10.000 | 0.500 | 0.500 | 10.00 | 0.500 |
| $V_{m,25}$ | Maximum catalytic capacity of Rubisco per unit leaf area at 25°C | $\mu$mol m$^{-2}$ s$^{-1}$ | 22.560 | 28.450 | 29.560 | 18.770 | 24.440 | 0.000 | 28.450 | 18.770 |
| $\beta$ | Initial photochemical efficiency | – | 0.029 | 0.029 | 0.029 | 0.029 | 0.029 | 0.000 | 0.029 | 0.029 |
| $\eta$ | Initial value of the slope of $CO_2$ response curve | mol m$^{-2}$ s$^{-1}$ | 0.069 | 0.040 | 0.026 | 0.024 | 0.069 | 0.000 | 0.040 | 0.024 |
| $m$ | Ball-Berry coefficient | – | 5.289 | 8.355 | 3.934 | 4.406 | 9.211 | 0.000 | 8.355 | 4.406 |
| $D_{min}$ | The threshold below which there is no vapor pressure constraint | kPa | 1.499 | 0.711 | 0.650 | 1.493 | 0.664 | 1.000 | 0.711 | 1.493 |
| $D_{max}$ | The threshold above which there is no assimilation | kPa | 6.500 | 3.500 | 5.199 | 5.797 | 5.188 | 6.500 | 3.500 | 5.797 |

[a] CRO: cropland, MIF: mixed forest, GRA: grassland, SHR: shrubland, WET: wetland, WAT: water body, IMP: impervious surface and BAR: bare land.

2.2.1. Energy Balance at Urban Land Surface

Based on the surface energy balance, the net radiation ($R_n$) can be balanced by the latent heat flux ($LE$), sensible heat flux ($H$) and ground heat flux ($G$). As for a biweekly or longer estimation, the $G$ is often negligible ($G \ll H + LE$), then the $H$ is given by

$$H = R_n - LE - G \approx R_n - LE \tag{3}$$

The net radiation at the surface is the sum of the net shortwave downward radiation and the net longwave downward radiation,

$$R_n = (1 - \alpha)SW + \left( LW - \epsilon \sigma T_a^4 \right) \tag{4}$$

where the shortwave downward radiation $SW$ (W m$^{-2}$), the longwave downward radiation $LW$ (W m$^{-2}$) and the surface air temperature $T_a$ (K) are from atmospheric forcing input data [10]. The shortwave albedo $\alpha$ (-) and the longwave emissivity $\epsilon$ (-) are from satellite-based estimations. $\sigma$ is the Stefan–Boltzmann constant ($5.67 \times 10^{-8}$ W m$^{-2}$ K$^{-4}$). The latent heat flux ($LE$, W m$^{-2}$) is calculated by $LE = \frac{1}{c}\lambda ET$ and

$$ET = E_c + E_s + E_i + E_u + E_w \tag{5}$$

where $\lambda = 2500 - 2.2(T_a - 273.15)$ is the latent heat of vaporization (kJ kg$^{-1}$) at $T_a$, and $c$ (=86.4) is a conversion factor for units from (MJ m$^{-2}$ d$^{-1}$) to (W m$^{-2}$). $ET$ is the evapotranspiration (mm d$^{-1}$) summed from the canopy transpiration ($E_c$) and soil evaporation ($E_s$), interception evaporation ($E_i$), impervious surface evaporation ($E_u$) and open-water evaporation ($E_w$).

2.2.2. Canopy Transpiration ($E_c$) and Soil Evaporation ($E_s$)

The transpiration at canopy scales ($E_c$) is coupled with the photosynthesis process ($A_{gs}$) via the dynamical modulation of the canopy stomatal conductance ($G_c$), and the soil evaporation ($E_s$) depends on absorbed energy flux and soil water deficit,

$$E_c = \frac{\varepsilon A_c + (\rho c_p / \gamma)DG_a}{\varepsilon + 1 + G_a/G_c} \tag{6}$$

$$E_s = \frac{f \varepsilon A_s}{\varepsilon + 1} \tag{7}$$

where the surface available energy ($A = R_n - G$) is divided into canopy absorbed energy ($A_c$) and soil absorbed energy ($A_s$), $A_c = (1 - \tau)A$ and $A_s = \tau A$, $\tau = exp(-k_A LAI)$, $k_A = 0.6$. $\varepsilon = \frac{\Delta}{\gamma}$, and $\Delta$ is the slope of the curve relating saturation water vapor pressure to temperature (kPa °C$^{-1}$). $\rho$ is the density of air (g m$^{-3}$); $c_p$ is the specific heat of air at constant pressure (J g$^{-1}$ °C$^{-1}$); $D$ is vapor pressure deficit (kPa); $G_a$ is the aerodynamic conductance (m s$^{-1}$); $G_c$ (m s$^{-1}$) is the canopy conductance; $f$ is a dimensionless variable that determines the water availability for soil evaporation.

The canopy stomatal conductance ($G_c$) is calculated by the photosynthesis process ($A_{gs}$) for each PFT with the constraint of $D$ at surface.

$$G_c = \frac{m A_{gs}}{C_a(1 + D/D_0)} \tag{8}$$

$$A_{gs} = \frac{P_1 C_a}{k(P_2 + P_4)} \left\{ kLAI + ln \frac{P_2 + P_3 + P_4}{P_2 + P_3 \exp(kLAI) + P_4} \right\} \tag{9}$$

$$V_m = \frac{V_{m,25} \exp[a(T - 25)]}{1 + \exp[b(T - 41)]} \tag{10}$$

where $P_1 = A_m \beta I_0 \eta$; $P_2 = A_m \beta I_0$; $P_3 = A_m \eta C_a$; $P_4 = \beta I_0 \eta C_a$; $A_m = 0.5Vm$. $I_0$ is the photosynthetically active radiation (PAR, in mol) from shortwave downward radiation. $C_a$ is the atmospheric $CO_2$ concentration (in ppm or mol mol$^{-1}$). $V_{m,25}$, $\beta$, $\eta$, $m$, $D_{min}$, $D_{max}$ and $D_0$ are the parameters (see Table 1) in the PML-V2.1 model.

Finally, the gross primary productivity ($GPP$) is calculated by

$$GPP = A_{gs} f_D \tag{11}$$

$$f_D = \begin{cases} 1, & D < D_{min} \\ \frac{D_{max} - D}{D_{max} - D_{min}}, & D_{min} < D < D_{max} \\ 0, & D > D_{max} \end{cases} \tag{12}$$

where $f_D$ is the $D$ constraint function; $D_{min}$ and $D_{max}$ are the parameters (see Table 1).

### 2.2.3. Interception Evaporation ($E_i$) by Canopy Vegetation

The rainfall interception evaporation ($E_i$) is calculated by the van Dijk model, which was developed by van Dijk and Bruijnzeel [19,20], who modified it from the original Gash model [21,22]. The modified Van Dijk model used in this study is described by

$$E_i = \begin{cases} f_V P & for\ P < P_{wet} \\ f_V P_{wet} + f_{ER}(P - P_{wet}) & for\ P \geq P_{wet} \end{cases} \tag{13}$$

with

$$f_V = 1 - exp\left(-\frac{LAI}{LAI_{ref}}\right) \tag{14}$$

$$P_{wet} = -ln\left(1 - \frac{f_{ER}}{f_V}\right)\frac{S_V}{f_{ER}} \tag{15}$$

$$f_{ER} = f_V F_{ER0} \tag{16}$$

$$S_V = S_{leaf} LAI \tag{17}$$

where $P$ is rainfall rate (mm d$^{-1}$), and $P_{wet}$ is reference threshold precipitation rate when the canopy is wet (mm d$^{-1}$). $f_V$ describes the fraction area covered by intercepting leaves, which is determined by the leaf area index ($LAI$) and reference $LAI$ ($LAI_{ref}$, see Table 1) for the specific vegetation types. $f_{ER}$ is the ratio of the average evaporation rate over the average rainfall intensity during storms, and $S_V$ is the canopy rainfall storage capacity (mm), which currently is parameterized as water storage in the leaf at the canopy level. The $f_{ER0}$ and $S_{leaf}$ are the parameters shown in Table 1.

### 2.2.4. Impervious Surface Evaporation ($E_u$)

Impervious surface evaporation ($E_u$) is calculated by the revised van Dijk model in this study,

$$E_u = \begin{cases} f_U P & for\ P < P_{wet} \\ f_U P_{wet} + f_{ER}(P - P_{wet}) & for\ P \geq P_{wet} \end{cases} \tag{18}$$

with

$$f_U = 1 - f_V - f_w \tag{19}$$

$$P_{wet} = -ln\left(1 - \frac{f_{ER}}{f_U}\right)\frac{S_U}{f_{ER}} \tag{20}$$

$$f_{ER} = f_U F_{ER0} \tag{21}$$

where $f_U$ describes the fractional area covered by impervious surface in urban ecosystems, which is the rest fraction of vegetation coverage ($f_V$) and water body covered fraction ($f_w$). $S_U$ is the impervious surface storage capacity (mm).

### 2.2.5. Open-Water Evaporation ($E_w$)

The open-water evaporation ($E_w$) in lakes, rivers and other water bodies for Northern China is parameterized on the basis of Dalton's Law [23],

$$E_w = 0.144(1 + 0.75U_{1.5})[D + \Delta(T_{1.5})(\alpha - 1)T_{1.5}] \tag{22}$$

where $U_{1.5}$ and $T_{1.5}$ is wind speed (m/s) and air temperature (°C) at 1.5 m height, respectively. $\Delta(T_{1.5})$ is the slope of the curve relating saturation water vapor pressure to temperature $T_{1.5}$. $\alpha - 1$ is a regulator coefficient for atmospheric stability.

## 3. Results

### 3.1. Validation of Estimated LAI and ET

The LAI estimated from the Sentinel-2-based NDVI was compared to the observed LAI for June 2018 in the Beijing Sponge City (Figure 2a). The observed LAI was measured within the region around 39.50–40.50° N, 115.40–116.10° E. The result shows that the Sentinel-2-based LAI has a high correlation with the observed values ($R^2$ = 0.74), indicating that the LAI at 10m resolution estimated from Sentinel-2 can be well applied to estimate ET for the Beijing Sponge City.

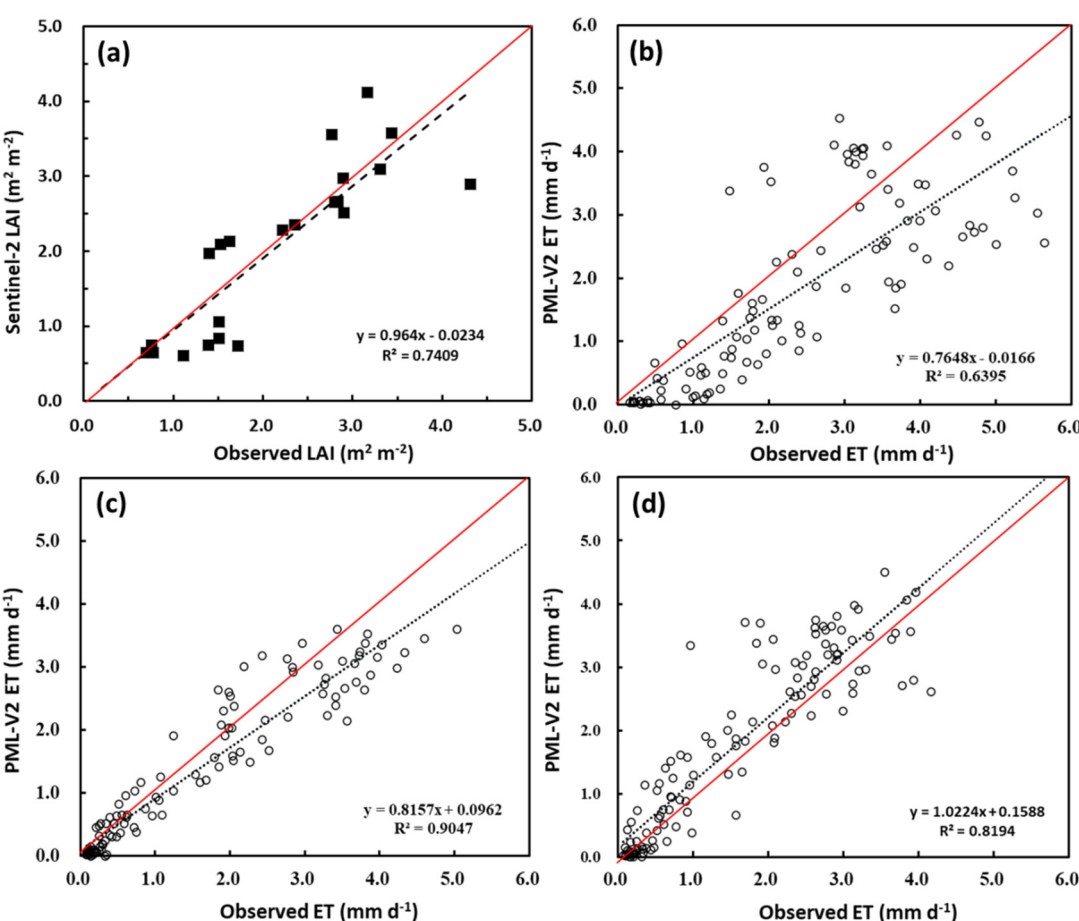

**Figure 2.** Validation of the Sentinel-2-retrieved LAI and the PML-V2-simulated ET. (**a**) The Sentinel-2-retrieved LAI compared to observed LAI for June 2018; (**b**–**d**) The PML-V2-simulated 8-day ET based on MODIS LAI compared to observed ET for the three flux tower sites at Daxing, Miyun and Guantao, respectively, in the Beijing area over 2008–2010.

We also validate the performance of PML-V2 in simulating ET in the Beijing region. Figure 2b–d shows the comparisons of PML-V2-estimated ET based on MODIS LAI with the observed ET over 2008–2010 from three field sites (i.e., Daxing, Miyun and Guantao)

which are located within the Beijing region. The result indicates that PML-V2 has satisfied performance in simulating ET for Beijing Sponge City with ($R^2$ = 0.64–0.90). Therefore, based on the above good performance in the Sentinel-2-estimated LAI and the PML-V2-estimated ET, we further evaluate NDVI, LAI, and ET and related variables at 10m resolution for Beijing Sponge City.

### 3.2. Land Use and Vegetation Information in Beijing Sponge City

By analyzing the Sentinel-2-derived 10m resolution LCC map in 2017, we find that the Beijing Sponge City project (Figure 3) covers ~1265 km$^2$ over the central Beijing city, China, including impervious surface buildings (59.27%), grasslands (26.08%), mixed forest (7.34%), croplands (5.10%), and wetlands and water bodies (~2%). Figure 3 presents fine spatial details of the urban ecosystem, such as clear patterns of lakes, rivers and streets. Most grasslands are parks, and fixed forests are mainly concentrated in northwestern Beijing Sponge City, while the eastern parts are croplands.

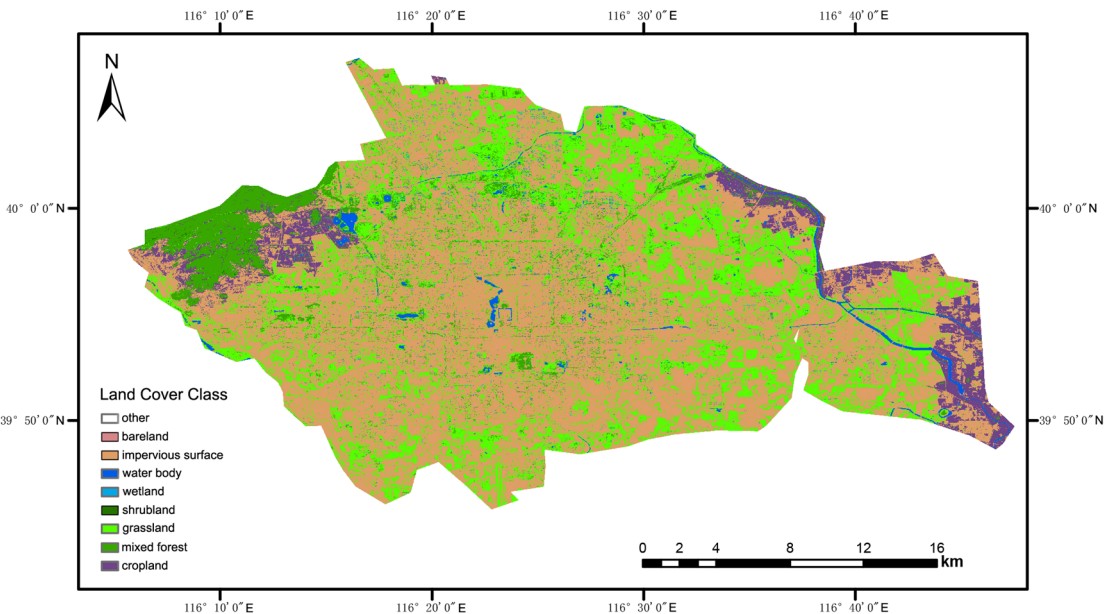

**Figure 3.** Land cover classification at 10m resolution for Beijing Sponge City. Land cover map for 2017 derived from the FROM-GLC10-based Sentinel-2.

We further analyze the NDVI and LAI for June 2018, which was composited from 10-day Sentinel-2 images in June 2018 in clear-sky conditions. We can see high spatial details in the NDVI from Figure 4a. The NDVI for lakes and rivers is ≤0.0, the impervious surfaces (e.g., large mansions and main streets) are 0.0–0.25, and grasslands and croplands are 0.1–0.5, while mixed forests and some parks with forest reserve have NDVI values of 0.5–0.7 (Figure 4a). A high NDVI indicates a high LAI in this study. Figure 4b shows the detailed pattern of LAI for different land cover classes for the sponge city area. As expected, the lakes and rivers have no LAI, and the impervious surfaces (e.g., large mansions and main streets) have only <1.0 m$^2$ m$^{-2}$ of LAI, but 1~2 m$^2$ m$^{-2}$ of LAI can be seen in many avenues with greenbelts. The mixed forests in northwestern Beijing Sponge City have LAI values ranging from 1 to 3 m$^2$ m$^{-2}$. Most grasslands and croplands have 1~2 m$^2$ m$^{-2}$ of LAI, but some parks with forest reserves show the highest values (3–8 m$^2$ m$^{-2}$) of LAI (Figure 4b).

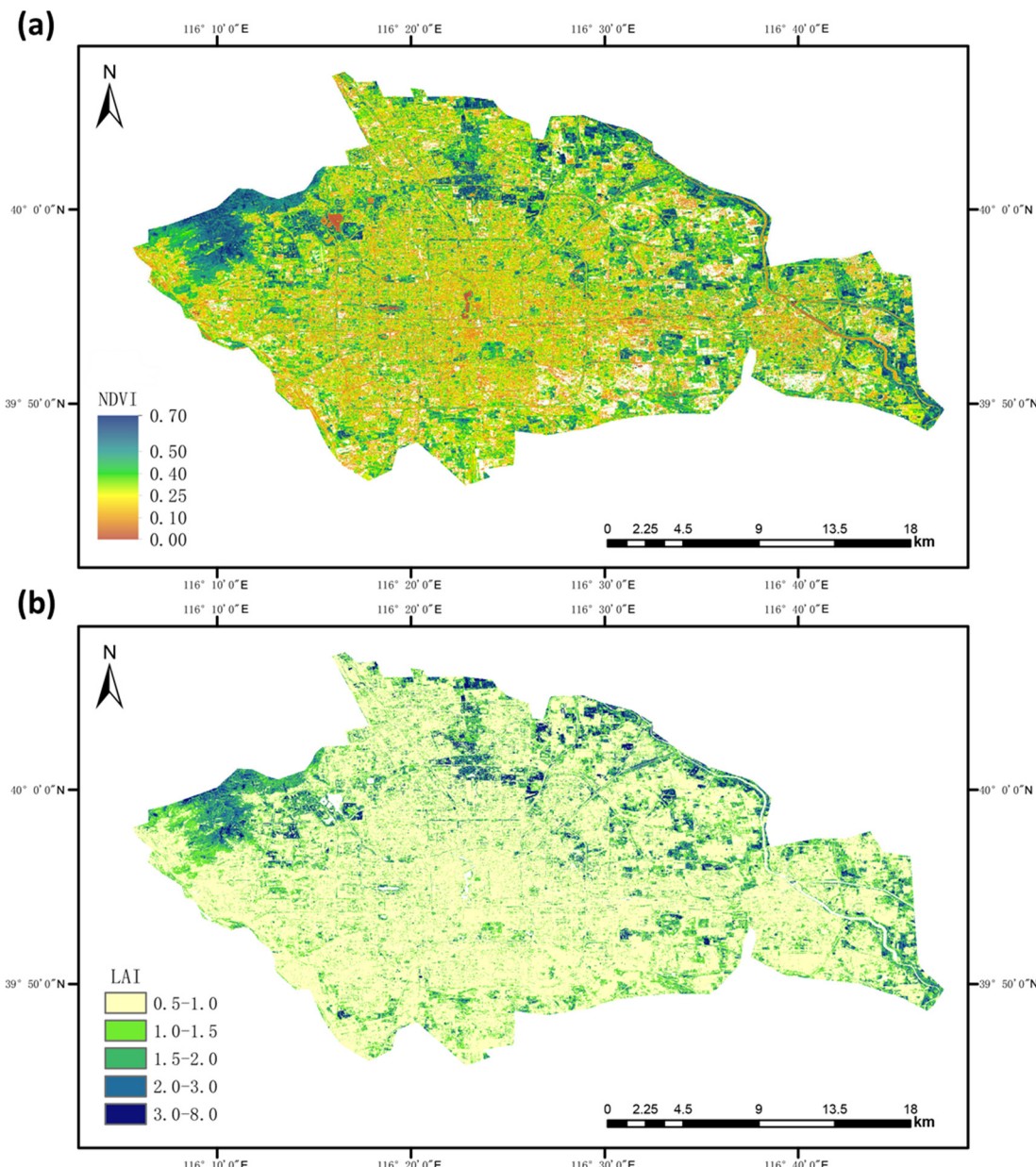

**Figure 4.** Vegetation index at 10m resolution for Beijing Sponge City. (**a**) NDVI and (**b**) LAI (m$^2$ m$^{-2}$) for June 2018 derived from the Sentinel-2.

### 3.3. ET and Related Variables in Beijing Sponge City

Based on the Sentinel-2-derived LCC and LAI data, we conducted daily simulations for June 2018 at 10m resolution using the PML-V2 model with daily climate forcing data from CMFD v1.0. Figure 5 presents the spatial patterns of simulated monthly ET and GPP averaged over the daily output for June 2018. Lakes and rivers have the highest ET ($\geq$8 mm d$^{-1}$) due to the full water supply for evaporation in Summer. There is no GPP in lakes and rivers as simulated by PML-V2. The vegetation production activities are strongest in mixed forests and croplands, with the GPP ranging from 8 to 16 gC m$^{-2}$ d$^{-1}$ (Figure 5b). In these mixed forests and croplands, the ET is also high (4–6 mm d$^{-1}$), where the plant transpiration (Ec) plays a dominant role with ratios of Ec to ET larger than 0.8 (Figure 6a). In addition, the impervious surfaces have very small ET (<2 mm d$^{-1}$) and GPP (<4 gC m$^{-2}$ d$^{-1}$), indicating both Ec and soil evaporation (Es) are very small in these areas. The grasslands have 2–4 mm d$^{-1}$ of ET and 4–8 gC m$^{-2}$ d$^{-1}$ of GPP in the sponge city (Figure 5), where the ratio Ec/ET are 0.5–0.7 (Figure 6a).

In summary, on average, for the whole sponge city, we find that the LAI in June 2018 is 0.83 m$^2$ m$^{-2}$; the ET is about 1.6 mm d$^{-1}$; the GPP is 2.8 gC m$^{-2}$ d$^{-1}$. Table 2 further gives the evaluation for different districts in the sponge city. It shows that the central areas (i.e., Xicheng and Dongcheng districts) have the smallest LAI (0.66–0.7 m$^2$ m$^{-2}$), ET (~1.61 mm d$^{-1}$) and GPP (2.36–2.44 gC m$^{-2}$ d$^{-1}$), while the western areas (i.e., Shijingshan and Haidian districts) have the highest LAI (0.93–1.05 m$^2$ m$^{-2}$), ET (~1.65 mm d$^{-1}$) and GPP (3.10–3.53 gC m$^{-2}$ d$^{-1}$).

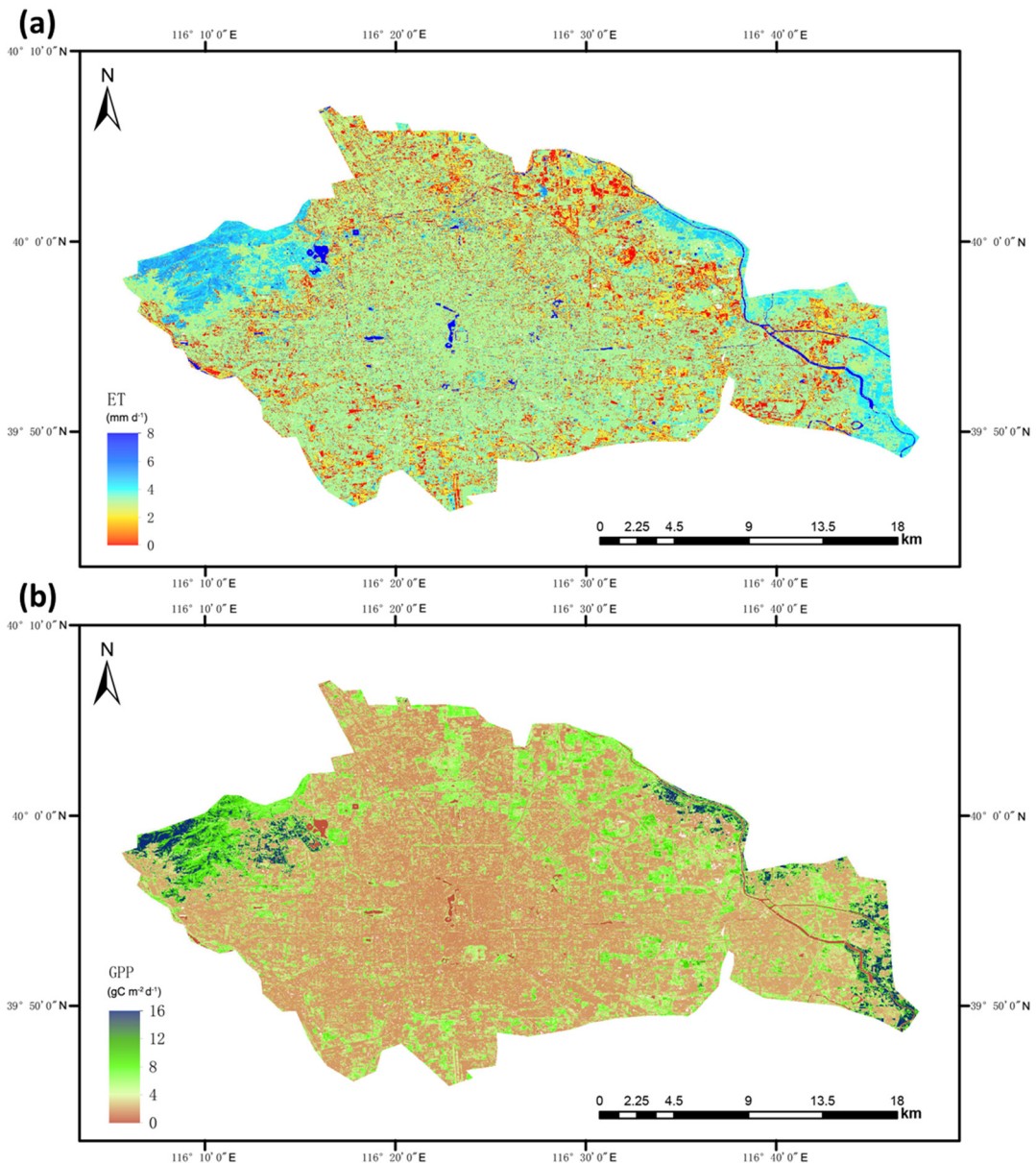

**Figure 5.** ET and GPP at 10m resolution for Beijing Sponge City estimated using the PML-V2 model and Sentinel-2 data. (**a**) Monthly ET (mm d$^{-1}$) in June 2018; (**b**) Monthly GPP (gC m$^{-2}$ d$^{-1}$) in June 2018.

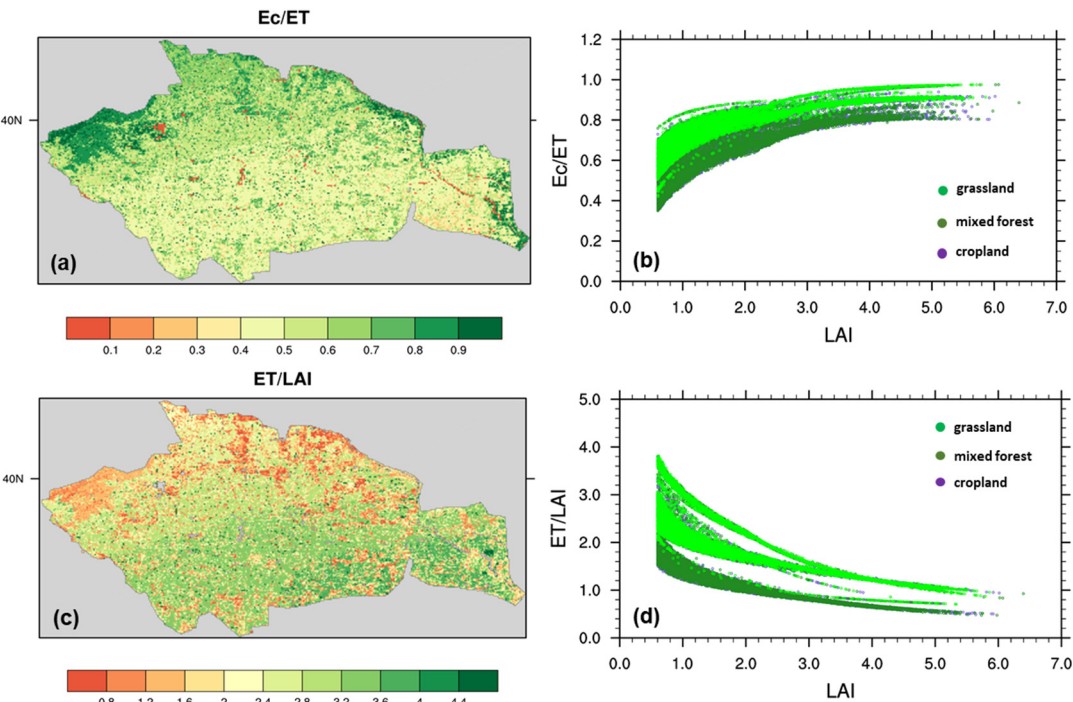

**Figure 6.** Spatial sensitivity of ET to LAI for the Beijing Sponge City. (**a**) Spatial pattern of the fraction of Ec to ET in June 2018; (**b**) Change in the fraction of Ec to ET with LAI; (**c**) Spatial pattern of the ratio of ET to LAI in June 2018; (**d**) Change in ratio of ET to LAI with LAI.

**Table 2.** The ecohydrological environment in different districts in Beijing Sponge City.

| District in Beijing Sponge City | LAI ($m^2$ $m^{-2}$) | ET (mm $d^{-1}$) | GPP (gC $m^{-2}$ $d^{-1}$) |
|---|---|---|---|
| Xicheng | 0.66 | 1.62 | 2.36 |
| Dongcheng | 0.70 | 1.60 | 2.44 |
| Shijingshan | 1.05 | 1.67 | 3.53 |
| Haidian | 0.93 | 1.64 | 3.10 |
| Chaoyang | 0.84 | 1.51 | 2.79 |
| Fengtai | 0.74 | 1.50 | 2.56 |
| Tongzhou | 0.86 | 1.66 | 3.01 |
| Overall Mean | 0.83 | 1.60 | 2.83 |

We further investigate how ET changes spatially with increasing LAI for June 2018. It is shown that the fraction Ec/ET increases with LAI for the three vegetation types (grassland, mixed forest and cropland) in the sponge city (Figure 6b). The Ec/ET for mixed forests increase from 0.4 (LAI = 0.5) to 0.8 (LAI > 3), while Ec/ET for grasslands and croplands show higher values, increasing from 0.6 (LAI = 0.5) to 0.9 (LAI > 3). The ratio ET/LAI represents the amount of water loss per unit LAI. In Figure 6c, we can find that ET/LAI for mixed forests and some grassland parks show the lowest ET/LAI (0.8–1.2), while impervious surfaces have the highest ET/LAI, with about 2-3 times larger values (2.4–3.6). The ET/LAI for the major vegetation types (grassland, mixed forest and cropland) decrease with LAI increase (Figure 6d); with LAI increasing from 0.5 to 5.0, the ET/LAI for mixed forests decrease from 2.0 to 0.6, and ET/LAI for grasslands and croplands from 3.0 to 1.0. This result indicates that grasslands and croplands have much higher water consumption per unit LAI than mixed forests.

Fractional contributions of other ET components to ET have been estimated (Figure 7). Soil evaporation (Es) contributes a relatively small fraction (≤0.2) due to a very small fraction of bare lands and large vegetation coverage in mixed forests, grasslands and

croplands in June 2018 in the Beijing Sponge City. The fraction of Ei/ET is small (≤0.1) in most LCC types but in grasslands is 0.1–0.2 (Figure 7b). According to a previous study, the city impervious surface could contribute less than 20% of ET [4]. Surprisingly, the impervious surface evaporation (Eu) for the Beijing Sponge City contributes 0.2–0.3 fractions to the ET in June 2018 in most impervious areas (Figure 7c). Finally, all ET from lakes and rivers are contributed by water body evaporation Ew (Figure 7d).

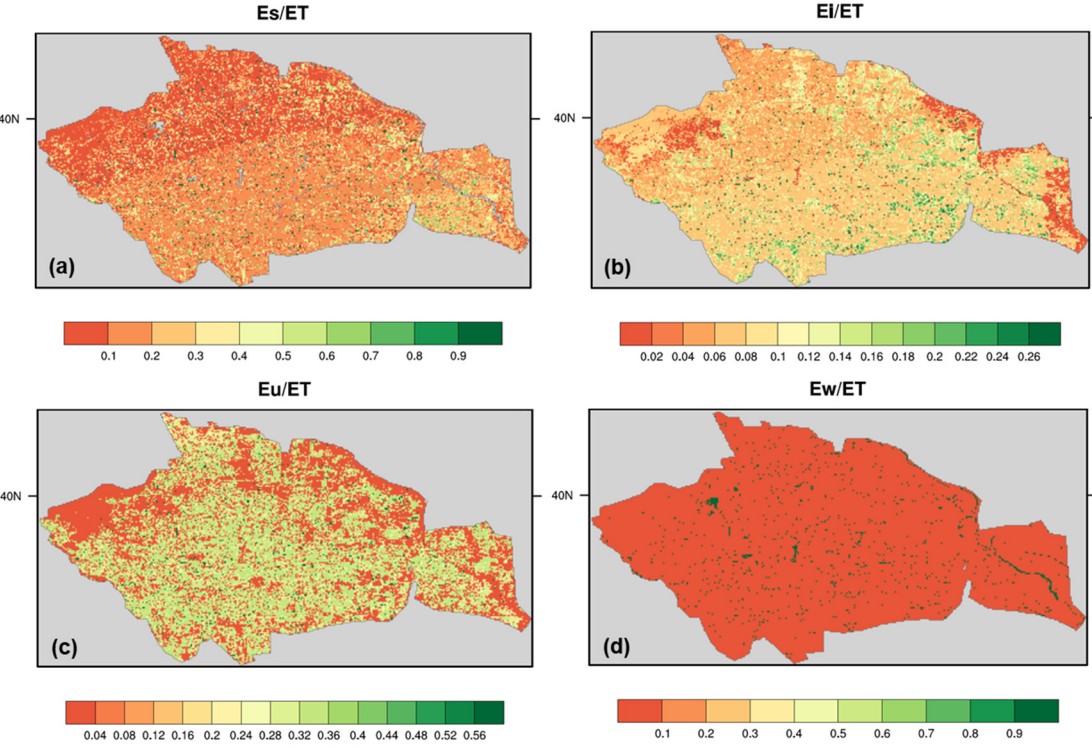

**Figure 7.** The fraction of Es, Ei, Eu and Ew to ET in June 2018 for the Beijing Sponge City. (**a**) Spatial pattern of fraction Es/ET; (**b**) Spatial pattern of fraction Ei/ET; (**c**) Spatial pattern of fraction Eu/ET; (**d**) Spatial pattern of fraction Ew/ET.

ET can be converted to latent heat flux (LE, in W m$^{-2}$) and plays an important role in regulating surface energy balance (Figure 8). For June 2018 in Beijing, the surface receives about 260–270 W m$^{-2}$ of shortwave radiation, but about half is reflected in the atmosphere with the net radiation (Rn) for impervious surface less than 130 W m$^{-2}$. For mixed forests and grasslands, the Rn is about 140–150 W m$^{-2}$, and the croplands and water bodies have a higher Rn of 170–180 W m$^{-2}$ (Figure 8b). Lakes and rivers have the highest LE (>250 W m$^{-2}$) but the smallest sensible heat flux (SH, <−60 W m$^{-2}$). The SH for mixed forests, forest parks and croplands are relatively small (−20 to 20 W m$^{-2}$), while both impervious surfaces and grasslands are high (40–60 W m$^{-2}$) (Figure 8d). This result indicates that the high surface air temperature (reflected by SH) in the summer of Beijing's central city is mostly contributed by impervious surfaces and grasslands.

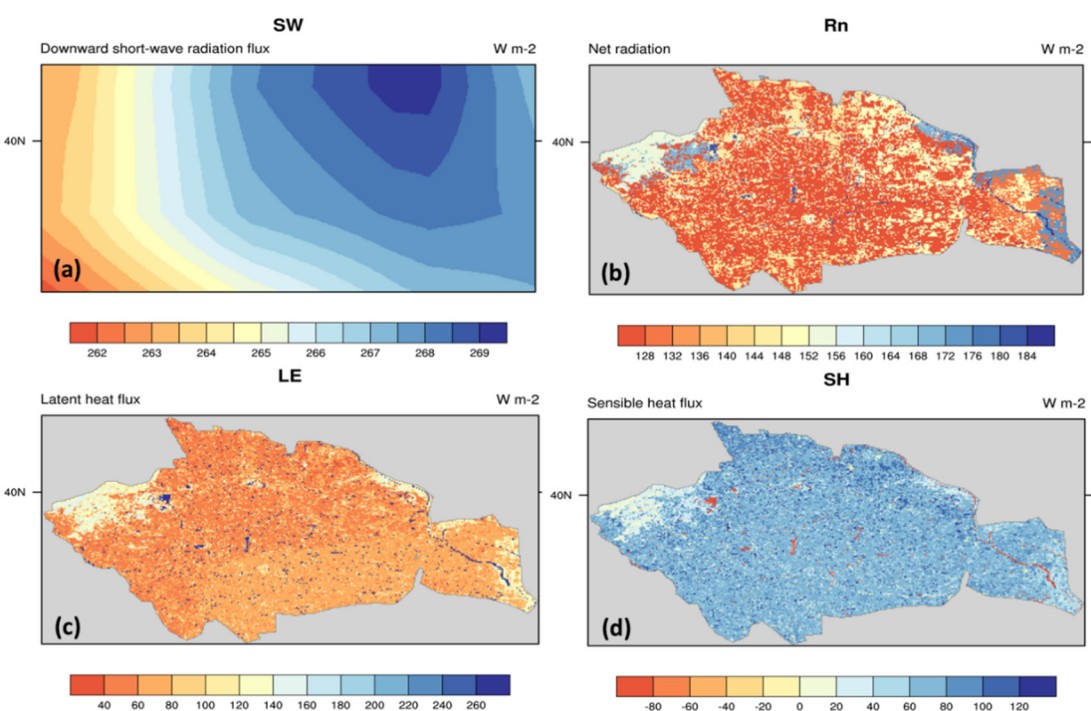

**Figure 8.** Energy fluxes in June 2018 for the Beijing Sponge City. (**a**) Downward shortwave radiation; (**b**) Surface net radiation; (**c**) Latent heat flux; (**d**) Sensible heat flux. Units are in W m$^{-2}$.

## 4. Discussion

In this study, we applied a water-carbon coupling model (PML-V2) with the use of Sentinel-2 LAI and land use and cover data and with surface meteorological forcing data as the input to estimate urban ET and its components at 10m resolution for the Beijing Sponge City. Our results indicate that the current vegetation coverage for the Beijing Sponge City is still at a low level (only with mean LAI < 1 m$^2$ m$^{-2}$ in June 2018), and the city gains relatively limited benefits from urban ecosystem services.

Eddy covariance measurements to study evaporation from urban ecosystems [24,25] generally helped us to understand the combined evaporation from all land cover types, lacking the capability to divide individual contributions from such as impervious surfaces (roofs, roads and plazas, etc.) and vegetated areas (bare soil, forests, grasslands and croplands, etc.). For different land cover classes, by using an advanced water-carbon coupling ET model driven by Sentinel-2 LAI, we find that lakes and rivers have the highest ET ($\geq$8 mm d$^{-1}$), followed by mixed forests and croplands (ET is 4–6 mm d$^{-1}$ and LAI is 2–3 m$^2$ m$^{-2}$), where the plant transpiration (Ec) plays the dominant role (>80%), then grasslands have 2–4 mm d$^{-1}$ of ET, where the LAI is 1~2 m$^2$ m$^{-2}$, while impervious surfaces have smallest ET (<2.0 mm d$^{-1}$). In most impervious areas, the impervious surface evaporation (Eu) contributes 20-30% of ET, which is larger than the estimate (18%) from previous studies [4]. We have shown that another 40–50% of ET in impervious areas are contributed by plant transpiration (Ec) due to the small fractional area covered by greenbelts with trees and grassland (LAI is 0.5–1.0 m$^2$ m$^{-2}$). This study did not consider water vapor conversion from human water use activities, which also contributes to the impervious evaporation from building indoor water use [25].

## 5. Conclusions

First of all, we show the good performances of the nonlinear regression model (Equation (2)) for estimating Sentinel-2 LAI based on the strong exponential relationship between NDVI and LAI and the PML-V2.1 model of estimating ET and GPP at 10m resolution using Sentinel-2 LAI and land cover map. This Sentinel-2-based technology using the

PML-V2.1 model with surface meteorological forcing data as the input for estimating ET provides a new framework to evaluate city-level hydrological dynamics in urban ecosystems. Secondly, we find that plant transpiration from greenbelts with trees and grassland play an important role in most impervious areas for Beijing Sponge City. Thirdly, mixed forests, forest parks and croplands due to their high ET have much smaller surface heat contribution than the impervious and grasslands, providing better ecosystem services (e.g., cooling) for the sponge city.

**Author Contributions:** Conceptualization, X.Z.; methodology, X.Z.; software, X.Z. and P.S.; validation, X.Z. and P.S.; formal analysis, X.Z.; investigation, X.Z.; resources, X.Z.; data curation, X.Z.; writing—original draft preparation, X.Z.; writing—review and editing, X.Z. and P.S.; visualization, X.Z. and P.S.; supervision, X.Z.; project administration, X.Z.; funding acquisition, X.Z. All authors have read and agreed to the published version of the manuscript.

**Funding:** This research was funded by the Beijing Municipal Natural Science Foundation (grant number 8212017), the National Natural Science Foundation of China (grant number 42001019) and the Shanghai Sailing Program (grant number 19YF1413100). The APC was funded by the Beijing Municipal Natural Science Foundation (grant number 8212017).

**Data Availability Statement:** The resulting dataset in this study is available at https://doi.org/10.6084/m9.figshare.14387630.v2 (accessed on 8 April 2021). More data and codes can be accessible from the corresponding author upon request.

**Acknowledgments:** We acknowledge the Beijing Water Science and Technology Institute (BWSTI) for providing technical and financial supports to this work. We thank the three anonymous reviewers and editors for their constructive suggestions and comments.

**Conflicts of Interest:** The authors declare no conflict of interest.

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
