# Peer review of "Estimating Urban Evapotranspiration at 10m Resolution Using Vegetation Information from Sentinel-2: A Case Study for the Beijing Sponge City"

_remotesensing, doi:10.3390/rs13112048_

Round 1

Reviewer 1 Report

The authors have addressed all my concerns and corrections, the manuscript can be published in the present form 

Reviewer 2 Report

The reviewer is very grateful to the authors for the reworking of the paper. The paper has become much better.

This manuscript is a resubmission of an earlier submission. The following is a list of the peer review reports and author responses from that submission.

Round 1

Reviewer 1 Report

The manuscript "Estimating urban evapotranspiration at 10m resolution using 2 satellite Sentinel-2: A case study for the Beijing Sponge City", by Zhang and Song, propose an estimation of the evapotranspiration in urban ecosystems using recent developments of remote sensing technology (Sentinel-2 products). The manuscript presents the methodology used in a clear manner and is well-written. There are only minor concerns and typos, that have been highlighted in the attached file. My recommendation is publish it after a minor revision

Author Response

Author's Reply to the Review Report (Reviewer 1)

Comments: The manuscript "Estimating urban evapotranspiration at 10m resolution using 2 satellite Sentinel-2: A case study for the Beijing Sponge City", by Zhang and Song, propose an estimation of the evapotranspiration in urban ecosystems using recent developments of remote sensing technology (Sentinel-2 products). The manuscript presents the methodology used in a clear manner and is well-written. There are only minor concerns and typos, that have been highlighted in the attached file. My recommendation is publish it after a minor revision.

Response: Thank you for reviewing this study and providing detailed comments for the manuscript. We have revised all the minor comments, please check highlighted texts in the revised manuscript.

Reviewer 2 Report

The reviewer is really grateful to the authors for the outstanding and so vital nowadays estimation of evapotranspiration in urban ecosystems. The authors are using recent developments of remote sensing technology to estimate leaf area index (LAI) from Sentinel-2 based Normalized Difference Vegetation Index (NDVI) using the NDVI LAI nonlinear relationship. By applying Sentinel-2-based LAI and land cover classification (LCC) to a carbon-water coupling model (PML-V2.1), the authors for the first time estimate monthly ET at 10m×10m resolution for the Beijing Sponge City. Results show that for the whole sponge city during June 2018, the LAI, ET and GPP are 0.83 m2 m-2,1.6 mm d-1 and 2.8 gC m-2 d-1 respectively.
However there are some minor inconveniences in the text formatting such as for example absence of upper indexes in the abstract (lines 16-21) and or non-uniform DOI indexes appointing in the reference list (somewhere is just DOI number, somewhere is the whole URL) such as for example in the lines 332, 345, 375, 378, 382… And it seems, that may be space after comma is needed in the references like in line 294 [24,25] for example...

Author Response

Author's Reply to the Review Report (Reviewer 2)

Comments: The reviewer is really grateful to the authors for the outstanding and so vital nowadays estimation of evapotranspiration in urban ecosystems. The authors are using recent developments of remote sensing technology to estimate leaf area index (LAI) from Sentinel-2 based Normalized Difference Vegetation Index (NDVI) using the NDVI LAI nonlinear relationship. By applying Sentinel-2-based LAI and land cover classification (LCC) to a carbon-water coupling model (PML-V2.1), the authors for the first time estimate monthly ET at 10m×10m resolution for the Beijing Sponge City. Results show that for the whole sponge city during June 2018, the LAI, ET and GPP are 0.83 m2 m-2,1.6 mm d-1 and 2.8 gC m-2 d-1 respectively.

However there are some minor inconveniences in the text formatting such as for example absence of upper indexes in the abstract (lines 16-21) and or non-uniform DOI indexes appointing in the reference list (somewhere is just DOI number, somewhere is the whole URL) such as for example in the lines 332, 345, 375, 378, 382… And it seems, that may be space after comma is needed in the references like in line 294 [24,25] for example...

Response: Thanks for highlighting the importance of this work. We have revised the minor comment, please check highlighted texts or tracked changes in the revised manuscript.

Reviewer 3 Report

A very good and articulated paper reporting a case study on the estimation of urban Evapotranspiration using Sentinel-2.

The manuscript is very clear in the introduction (please see one minor comment) and extremely detailed in the presentation fo the methodology: all subchapters are very clear and reporting the physical processes underpinned. I really like this presentation of the methodology which indicates clearly all the steps to reach the results. The results are very well organised and reported, and are extremely credible. The discussion is well focussed and the conclusions adeherent to the results obtained. References and figures appropriate to the level of the manuscript.

Comment (one minor): in the introduction at line 54 the authors report "to reduce stress on water supply..." could be interesting to know the total water demand of Beijing Sponge city.

Author Response

Author's Reply to the Review Report (Reviewer 3)

Comments: A very good and articulated paper reporting a case study on the estimation of urban Evapotranspiration using Sentinel-2.

The manuscript is very clear in the introduction (please see one minor comment) and extremely detailed in the presentation fo the methodology: all subchapters are very clear and reporting the physical processes underpinned. I really like this presentation of the methodology which indicates clearly all the steps to reach the results. The results are very well organised and reported, and are extremely credible. The discussion is well focussed and the conclusions adeherent to the results obtained. References and figures appropriate to the level of the manuscript.

Comment (one minor): in the introduction at line 54 the authors report "to reduce stress on water supply..." could be interesting to know the total water demand of Beijing Sponge city.

Response: Thanks for highlighting the importance of this work and for the compliment of the writing. We have revised the minor comment, please check line 57 in the revised manuscript.